# Measuring the Urban Land Surface Temperature Variations Under Zhengzhou City Expansion Using Landsat-Like Data

**Haibo Yang** [1], **Chaofan Xi** [1], **Xincan Zhao** [2,*], **Penglei Mao** [3], **Zongmin Wang** [1], **Yong Shi** [1,4], **Tian He** [1] and **Zhenhong Li** [5]

[1] School of Water Conservancy Engineering, Zhengzhou University, Zhengzhou 450001, China; yanghb@zzu.edu.cn (H.Y.); xcf@gs.zzu.edu.cn (C.X.); zmwang@zzu.edu.cn (Z.W.); yongshi@zzu.edu.cn (Y.S.); he_t@zzu.edu.cn (T.H.)
[2] School of Information Engineering, Zhengzhou University, Zhengzhou 450001, China
[3] Power China ZhongNan Engineering Corporation limited, Changsha 410014, China; 02973@msdi.cn
[4] Institute of Geographic Sciences and Natural Resources Research, CAS. Beijing 100101, China
[5] School of Engineering, Newcastle University, Newcastle upon Tyne NE1 7RU, UK; zhenhong.li@newcastle.ac.uk
* Correspondence: iexczhao@zzu.edu.cn; Tel.: +86-371-67781533

**Abstract:** Satellite-based remote sensing technologies are utilized extensively to investigate urban thermal environment under rapid urban expansion. Current Moderate Resolution Imaging Spectroradiometer (MODIS) data are, however, unable to adequately represent the spatially detailed information because of its relatively coarser spatial resolution, while Landsat data cannot explore the temporally continued analysis due to the lower temporal resolution. Combining MODIS and Landsat data, "Landsat-like" data were generated by using the Flexible Spatiotemporal Data Fusion method (FSDAF) to measure land surface temperature (LST) variations, and Landsat-like data including Normalized Difference Vegetation Index (NDVI) and Normalized Difference Built Index (NDBI) were generated to analyze LST dynamic driving forces. Results show that (1) the estimated "Landsat-like" data are capable of measuring the LST variations; (2) with the urban expansion from 2013 to 2016, LST increases ranging from 1.80 °C to 3.92 °C were detected in areas where the impervious surface area (ISA) increased, while LST decreases ranging from −3.52 °C to −0.70 °C were detected in areas where ISA decreased; (3) LST has a significant negative correlation with the NDVI and a strong positive correlation with NDBI in summer. Our findings can provide information useful for mitigating undesirable thermal conditions and for long-term urban thermal environmental management.

**Keywords:** land surface temperature (LST); spatiotemporal fusion; MODIS; urbanization; driving forces

## 1. Introduction

Worldwide urban expansions are developing at various spatiotemporal scales under the rapid population and economy growth [1–4]. The World Urbanization Prospects 2014 Revision report stated that 54% of the world's population lived in urban regions in 2014 and the number would be projected to grow to 66% in 2050 [5]. Inevitably, urban expansion caused land use and land cover change and brought about many environment or ecological system problems [6]. Especially, the coupling of rapid urban expansion and climate warming has increased the heat stress in many mega cities [7]. A great deal of natural vegetation and farmland was replaced by man-made impervious surface area, leading to dramatic change of land surface temperature (LST) and local climate [8–10]. LST is an indicator

showing the biosphere–land–ocean–atmosphere interactions and surface energy balance, and it can reflect the thermal response to urban canopy, building height, surface coverage, anthropogenic heat, and energy consumption [11–13].

Traditionally, near-surface air temperatures as a proxy of in-situ LST can be accurately measured by using in-situ thermometers but are limited in station-based observation [14,15]. With the development of earth observation technologies, a range of remotely sensed datasets have been used to characterize LST and to investigate thermal variations across urban environments [16–18]. Satellite thermal infrared (TIR) LST, theoretically based on the radiative transfer equation, has been widely studied focusing on single-channel, split-window, and multi-channel algorithms for solving emissivity and atmospheric effects [19–21]. For example, Landsat and Moderate Resolution Imaging Spectroradiometer (MODIS) data are intensively used on different spatiotemporal scale LST studies [22–31] (Table 1). Landsat data with a higher spatial resolution of 60 m or 100 m in the thermal band are more suitable for detailed investigation of LST, however, Landsat data has lower temporal resolution (16 days) [32–36]. MODIS data, with higher temporal resolution (twice a day), are most popular in continuous studies satisfying a high temporal resolution time scale [37,38]. The low spatial resolution of MODIS scans can only be used to coarse-scale research. It is insufficient for producing detailed descriptions of LST variation or for analyzing thermal driving forces [39]. Similarly, the temporally sparse resolution of Landsat data cannot produce the short-interval datasets necessary to investigate LST dynamics [40].

Both MODIS and Landsat data have their strengths and limitations. To obtain simultaneously the higher spatiotemporal resolution data, data fusion methods have been used to deal with these spatiotemporal resolution problems by generating the fusion LST data of Landsat and MODIS, satisfying the need of long-term and fine-scale regional thermal environment research [41–44]. For example, by utilizing the Spatial and Temporal Adaptive Reflectance Fusion Model (STARFM), Shen et al. [45] obtained "Landsat-like" LST datasets from 1988 to 2013 and analyzed the thermal mechanism of the Wuhan city urban heat island. Huang et al. [46] proposed a spatiotemporal fusion model based on a bilateral filter method (STBFM) and testified its higher precision compared with the STARFM method. Weng et al. [47] developed a new fusion algorithm to predict daily LST. Wu et al. [48] put forward the spatiotemporal integrated temperature fusion model (STITFM) to integrate data from multiple sensors with flexibility. Besides the above three methods (STARFM, STBFM, and STITFM), a Flexible Spatiotemporal Data Fusion method (FSDAF) [49], which has a strong advantage in generating high spatiotemporal resolution data due to high calculation efficiency and minimum data requirements, was adopted by Zhang et al. [50] to generate monthly time series LST data.

Although there is different research focusing on the new data fusion algorithms to generate higher spatial and temporal resolution land surface factors, there are still few documents to indicate simultaneously the LST dynamics and the driving forces by combining the Landsat-like data Normalized Difference Vegetation Index (NDVI) and Normalized Difference Vegetation Index (NDBI). Zhengzhou is a provincial city in the Central plains of China. It has the highest urban population density in mainland of China and is a major economic center in the middle of China. It is currently undergoing a great-leap-forward development [51]. Moreover, the urban built-up area has increased rapidly, by 65.83% over the last decade. There is, therefore, an urgent requirement to quantitatively evaluate urban thermal environmental change and explore the principal driving mechanisms. The purpose of this study includes: (i) measuring urban land surface temperature dynamics under fast urban expansion by using Landsat-like LST, and (ii) analyzing LST driving forces combining Landsat-like NDVI and NDBI and other factors.

**Table 1.** The comparison of LST data.

| LST Data Source | Spatial Resolution/ Temporal Resolution | Time Scale | Strength and Limitation | Reference |
|---|---|---|---|---|
| Landsat | 60 m (TM and ETM+) or 100 m (OLI-TIRS)/16 d | monthly | High spatial resolution, low frequency | Chen X, et al. [22] Sheng L, et al. [23] |
| | | annual | | Xiong Y, et al. [24] Wang S, et al. [25] |
| MODIS | 1000 m/1 d | daily | High frequency, low spatial resolution | Li X, et al. [26] Sun L, et al. [27] |
| | | monthly | | Jose L, et al. [28] Williamson S, et al. [29] |
| | | annual | | Haynes M, et al. [30] Eleftheriou D, et al. [31] |
| Fusion data of Landsat and MODIS | 60 m (TM and ETM+) or 100 m (OLI-TIRS)/1 d | daily | High spatial resolution, high frequency | Huang B, et al. [46] Weng Q, et al. [47] |
| | | monthly | | Zhang L, et al. [50] |

LST: land surface temperature; MODIS: Moderate Resolution Imaging Spectroradiometer; TM: Thematic Mapper; ETM+: Enhanced Thematic Mapper Plus; OLI-TRIS: Operational Land Imager/Thermal Infrared.

## 2. Materials and Methods

*2.1. Materials*

The study area shown in Figure 1 is the Zhengzhou, located from 113°26′ to 113°51′ E, and 34°35′ to 34°57′ N. It comprises five municipal zones: Erqi (EQ), Zhongyuan (ZY), Huiji (HJ), Jinshui (JS), and Guancheng (GC). The built-up area of this conurbation expanded significantly, from 328.01 km² in 2011 to 443.04 km² in 2016 according to the dataset of National Bureau of Statistics of the People's Republic of China.

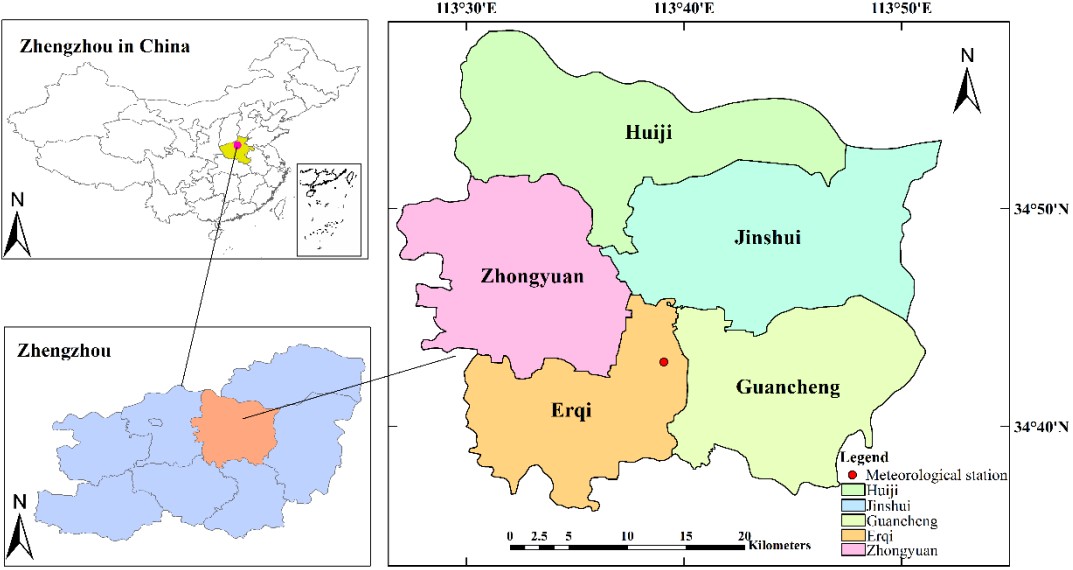

**Figure 1.** Map of the study area showing five municipal zones and meteorological station.

Data used in this paper mainly include two parts (Table S1): (1) four periods of data representing summer (28 June 2013 and 20 June 2016) and winter (22 January 2014 and 22 January 2017), and (2) whole years of data including daily and monthly MODIS data (from March 2013 to February 2014). Landsat Enhanced Thematic Mapper Plus (ETM+) and Operational Land Imager/Thermal Infrared Sensor (OLI-TRIS) (https://earthexplorer.usgs.gov) have a resolution of 60/100 m. MODIS data include Daily MODIS LST (MOD11A1), 8-day maximum synthetic MODIS LST (MOD11A2), and 16-Day Vegetation Indices MODIS NDVI (MOD13Q1) from the NASA website (https://modis.gsfc.nasa.gov). Meteorological data (including air pressure, temperature, and relative humidity) were obtained from the Chinese National Meteorological Information Center (http://data.cma.cn). Vector map dataset and Digital Surface Model (DSM) with 30 m resolution were obtained from the Geographical Information Monitoring Cloud Platform (http://www.dsac.cn).

*2.2. Methods*

2.2.1. Data Preprocessing and Image Classification

The preprocessing of ETM+/OLI-TRIS images (Level 1) includes radiometric calibration and atmospheric correction by using the ENVI5.1 software package. First, the thermal infrared band was corrected with "Thermal Atmospheric Correction" tool in ENVI5.1, and then it was resampled to 30 m by using the bilinear method. Next, for optical bands, atmospheric correction was performed by using the FLASSH function. Finally, the digital number (DN) of the thermal band was validated into the radiance, while the DN of the optical band was validated into reflectance. Furthermore, for solving the problem of missing pixels in Landsat ETM + SLC-off images taken since 2003, the nearest-neighborhood interpolation method was used to predict missing pixel values. In addition, a 120 m*120 m fishnet

(n = 8971) was generated to calculate the mean impervious surface area (ISA) ratio. The interval of ISA ratio was equally divided into 100 segments from zero to one, and corresponding mean LST in each interval was calculated.

The support vector machine (SVM) supervised classification method was used for image classification. As a machine learning method based on the statistical learning theory, it has a strong precision level [52–54]. The land use includes four types, impervious surface area (ISA), vegetation, water, and bare soil. Furthermore, we used the random sampling method to select more than 100 samples to assess the accuracy of the classification results.

### 2.2.2. Calculation of NDVI, NDBI, and LST

NDVI (Normalized Difference Vegetation Index) and NDBI (Normalized Difference Built-Up Index), as two typical land surface factors, can be calculated as follows:

$$\text{NDVI} = \frac{R_{NIR} - R_{RED}}{R_{NIR} + R_{RED}} \tag{1}$$

$$\text{NDBI} = \frac{R_{MIR} - R_{NIR}}{R_{MIR} + R_{NIR}} \tag{2}$$

where $R_{NIR}$ is the reflectance in the near infrared band, which is corresponding to Landsat ETM+ band4 (0.775–0.900 μm) and Landsat OLI-TIRS band5 (0.845–0.885 μm). $R_{RED}$ represents the reflectance in visible red band corresponding to ETM+ band 3 (0.630–0.690 μm) and OLI-TIRS band4 (0.630–0.680 μm), and $R_{MIR}$ represents the reflectance of middle infrared band corresponding to ETM+ band 5 (1.550–1.750 μm) and OLI-TIRS band6 (1.560–1.651 μm).

Landsat LST calculation is based on the radiative transfer equation:

$$B(F_S) = \left[L_\lambda - L^\uparrow - \tau(1 - \varepsilon)L_\downarrow\right]/\tau\varepsilon \tag{3}$$

where $B(F_S)$ represents the surface radiance; $L_\lambda$ is the spectral radiance at the sensor's aperture in $W\cdot m^{-2}\cdot sr^{-1}\cdot \mu m^{-1}$; $F_S$ stands for real land surface temperature (K); $\tau$ is the atmospheric transmittance; and $L^\uparrow$ and $L_\downarrow$ are the upward and downward atmospheric thermal radiance, respectively. The three parameters were estimated using image meta information and meteorological data as well as the atmospheric correction parameter calculator [55,56] provided by NASA Website; $\varepsilon$ is the land surface spectral emissivity, which is critical for accurate calculation of LST.

In this study, $\varepsilon$ was calculated based on NDVI threshold method to differentiate types of land use/land cover (LULC). When NDVI < 0, the LULC was mainly water, but when 0 ≤ NDVI < 0.15, LULC was dominated by urban impervious areas and bare soil [57]. When NDVI ≥ 0.727, it was regarded as full of the vegetation coverage area [57]. However, when 0.15 ≤ NDVI < 0.727, LULC became a mixed object type [58]. Therefore, $\varepsilon$ was calculated with Equation (4):

$$\varepsilon = \begin{cases} 0.9925, NDVI < 0 \\ 0.923, 0 \leq NDVI < 0.15 \\ 1.0094 + 0.047 \ln(NDVI), 0.15 \leq NDVI \leq 0.727 \\ 0.986, NDVI > 0.727 \end{cases} \tag{4}$$

Finally, $C_S$ (°C) corresponding to the real LST was obtained with the Planck formula:

$$C_S = \frac{K_2}{\ln\left[\frac{K_1}{B(F_S)} + 1\right]} - 273.15 \tag{5}$$

The values of $K_1$ and $K_2$ are associated with the effective wavelength of the thermal band of each sensor [59]. For the ETM+ image, $K_1$ is 666.09 $W/(m^2{}^*\mu m^*sr)$ and $K_2$ is 1282.71 K, but for the OLI-TIRS image, $K_1$ is 774.89 $W/(m^2{}^*\mu m^*sr)$ and $K_2$ is 1321.08 K.

### 2.2.3. The Flexible Spatiotemporal Data Fusion Method

By combining the advantages of various spatiotemporal data fusion models, the Flexible Spatiotemporal Data Fusion Method (FSDAF) [49] ensures the minimum input requirement of data sources while capturing the features of both gradual and abrupt changes. The MODIS LST data (1000 m resolution) were resampled to 30 m resolution based on the nearest-neighbor algorithm method. Additionally, all input images needed to be cropped into the same area with the predetermined number of rows and columns before performing the spatiotemporal fusion method. The input data included two parts: (1) a pair of Landsat LST and MODIS LST data at $T_1$, which were used to estimate the spatial differences between fine-resolution pixels and coarse-resolution pixels; (2) one MODIS LST at $T_2$, which was used as the spatial feature reference and applied to calculate the time differences between $T_1$ and $T_2$ for predicting fine-resolution LST at $T_2$. The implementation process was divided into six steps: (1) classify Landsat LST at time $T_1$; (2) estimate the temporal changes taking place for each class of coarse-resolution MODIS LST from $T_1$ to $T_2$; (3) predict the fine-resolution LST at $T_2$ based on predicted temporal changes and calculate pixel residuals of MODIS LST; (4) use the thin plate spline (TPS) interpolation function to predict the high-spatial-resolution LST based on the MODIS LST at $T_2$; (5) allocate the residuals to predicted high-spatial-resolution LST with the TPS interpolation function; and (6) generate final fine resolution "Landsat-like" LST at $T_2$ based on the weights of pixels in the moving windows, which are assigned by nearest-neighborhood information. The calculation process is as shown in (6)–(12):

$$\Delta R_{high}\left(x_{ij}, y_{ij}, b\right) = \varepsilon_{high}\left(x_{ij}, y_{ij}, b\right) + \Delta R_{high}(a, b) \tag{6}$$

$$\widetilde{R}_{high2}\left(x_{ij}, y_{ij}, b\right) = R_{high1}\left(x_{ij}, y_{ij}, b\right) + \sum_{k=1}^{n}\left[\omega_k \times \Delta R(x_k, y_k, b)\right] \tag{7}$$

where $\widetilde{R}_{high2}\left(x_{ij}, y_{ij}, b\right)$ represents the DN values of fine spatial resolution pixels; $R_{high1}\left(x_{ij}, y_{ij}, b\right)$ is Landsat pixel values; $\Delta R(x_k, y_k, b)$ is the change of spatial resolution from $T_1$ to $T_2$; $\omega_k$ is the weight; $\Delta R_{high}(a, b)$ represents the change of class a of high spatial resolution data on band b (represents LST, NDVI, and NDBI data) from $T_1$ to $T_2$; $\partial_{high}\left(x_{ij}, y_{ij}, b\right)$ is the residual, which is allocated to the high spatial resolution pixel *j* from MODIS LST pixel *i*.

The calculation of weight was carried out with reference to previous studies [41], and the TPS function that guides the residual distribution was conducted as shown in Equations (8)–(12):

$$\partial_{high}\left(x_{ij}, y_{ij}, b\right) = m\partial\left(x_{ij}, y_{ij}, b\right) \times W\left(x_{ij}, y_{ij}, b\right) \tag{8}$$

$$\partial\left(x_{ij}, y_{ij}, b\right) = \Delta R_{low}\left(x_{ij}, y_{ij}, b\right) - \frac{1}{m}\left[\sum_{j=1}^{m} R_{high2}{}^{TP}\left(x_{ij}, y_{ij}, b\right) - \sum_{j=1}^{m} R_{high1}\left(x_{ij}, y_{ij}, b\right)\right] \tag{9}$$

$$CW\left(x_{ij}, y_{ij}, b\right) = E_{h0}\left(x_{ij}, y_{ij}, b\right) + \partial\left(x_{ij}, y_{ij}, b\right)\left[1 - HI\left(x_{ij}, y_{ij}\right)\right] \tag{10}$$

$$E_{h0}\left(x_{ij}, y_{ij}, b\right) = R_{high2}{}^{SP}\left(x_{ij}, y_{ij}, b\right) - R_{high2}{}^{TP}\left(x_{ij}, y_{ij}, b\right) \tag{11}$$

$$R_{high2}{}^{SP}\left(x_{ij}, y_{ij}, b\right) = f_{TPS-b}\left(x_{ij}, y_{ij}\right) \tag{12}$$

where m is the number of sub-pixels in MODIS LST; $\partial\left(x_{ij}, y_{ij}, b\right)$ represents the residual values between Landsat LST pixels and the fine-resolution LST pixels predicted based on the temporal changes; $\Delta R_{low}\left(x_{ij}, y_{ij}, b\right)$ represents the pixel value changes of band b in MODIS LST from $T_1$ to $T_2$; $R_{high2}{}^{TP}\left(x_{ij}, y_{ij}, b\right)$ and $R_{high2}{}^{SP}\left(x_{ij}, y_{ij}, b\right)$ are high-spatial-resolution LST pixel values at $T_2$ based on temporal changes and optimized TPS interpolation function parameters, respectively, and $E_{h0}\left(x_{ij}, y_{ij}, b\right)$ is the difference between two types of LST pixels; $CW\left(x_{ij}, y_{ij}, b\right)$ represents the weights of

guiding residual allocation; $W(x_{ij}, y_{ij}, b)$ is the normalized weight of $CW(x_{ij}, y_{ij}, b)$; $HI$ is homogeneous coefficient; and $f_{TPS-b}(x_{ij}, y_{ij})$ is the TPS function corresponding to band b.

To better understand the procedures of how to generate the Landsat-like LST by using FSDAF, a workflow is shown in Figure 2.

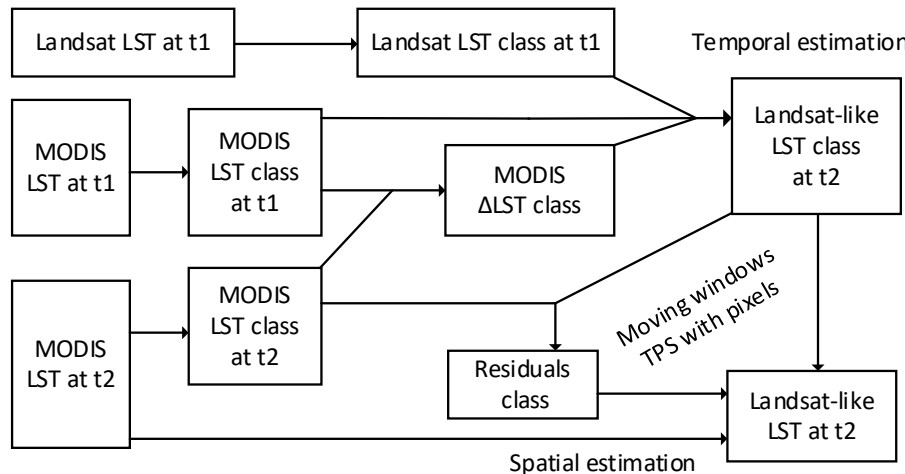

**Figure 2.** Flowchart of the Landsat-like LST generation procedure.

By using the ENVI IDL8.5 software package, the FSDAF model could be operated after many parameter settings were defined, including 30 pixels × 30 pixels window size, four classes of LST classification, and a similar pixel search threshold of 30. Table 2 shows the input and the output of the FSDAF method for predicting fine-resolution LST data. A pair of inputs (one MODIS LST and one Landsat LST) in T1 and another single input [MODIS LST data (as a reference of spatial feature)] in T2 were used in the FSDAF method to generate the output "Landsat-like" LST data in T2. To assure better fusion effects, the pair of input data should generally be cloud-free and close to the acquisition date of the single input data.

**Table 2.** The input and the output of the Flexible Spatiotemporal Data Fusion (FSDAF) method.

| Data | T1: The Pair of Inputs | | T2: Single Input | T2: Output |
|---|---|---|---|---|
| | **MODIS LST** | **Landsat LST** | **MODIS LST** | **Landsat-Like LST** |
| Verification | 4 June 2013 13 December 2013 | 4 June 2013 13 December 2013 | 28 June 2013 22 January 2014 | 28 June 2013 22 January 2014 |
| Prediction | 15 February 2017 | 15 February 2017 | 21 January 2017 | 21 January 2017 |

As for generating monthly-series "Landsat-like" LST data, there were two steps: (1) a pair of daily LST data (Landsat LST and MODIS LST <MOD11A1>) and 8-day MODIS LST (MOD11A2) were selected as inputs in the FSDAF model, and the output—8-day "Landsat-like" LST—could be predicted; and (2) four predicted 8-day "Landsat-like" LSTs in a month were integrated with maximum synthetic method to obtain monthly "Landsat-like" LST. If there was no Landsat image available in one month, the Landsat image of the previous or the next month was chosen, as well as the corresponding MOD11A1 in that date. If there were two Landsat images available in one month, four predicted 8-day "Landsat-like" LST were calculated according to the first and the second half month, respectively. In the study, for further discussing the FSDAF model significance on a serial analysis while data missing or cloud contaminating, the integrated monthly LST dynamics in a whole year were performed.

## 3. Results

### 3.1. Land-like LST Accuracy Assessment

The accuracy of Landsat-like LST was assessed by comparing it with MODIS LST and Landsat LST in three periods by using random points (n = 3000). On 28 June 2013 and 22 January 2014, Landsat-like LST data were generated for verification with Landsat LST on the same date (Figure 3). Because of the cloud coverage on 22 January 2017, by using MODIS LST on 21 January 2017, the predicated Landsat-like LST was generated and assessed (Figure 4). The root-mean-square errors (RMSEs) between the Landsat-like LST and the ETM+LST data were 2.46 °C and 1.73 °C, and the RMSEs were slightly smaller between Landsat-like LST and MODIS LST data, corresponding to 2.30 °C and 1.30 °C, respectively. In addition, the regression coefficients between Landsat-like LST and MODIS LST (0.96 and 1.01) were closer to one than those between Landsat-like LST and ETM+ LST (0.68 and 0.81) because the generation of Landsat-like LST was based on the MODIS LST. Figure 3 shows the predicted Landsat-like LST was highly similar to MODIS LST with a 0.976 regression coefficient ($R^2$ = 0.753). Spatially, all 30 m resolution Landsat-like LST had approximately the same "cold-hot" texture details distribution with Landsat LST and MODIS LST.

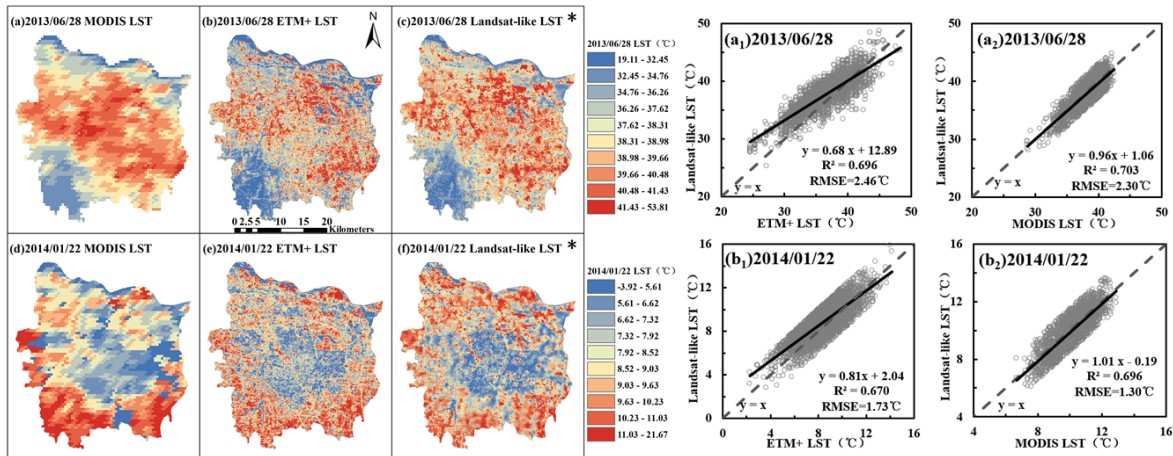

**Figure 3.** Comparison between MODIS/Landsat LST and "Landsat-like" LST in 28 June 2013 and 22 January 2014.

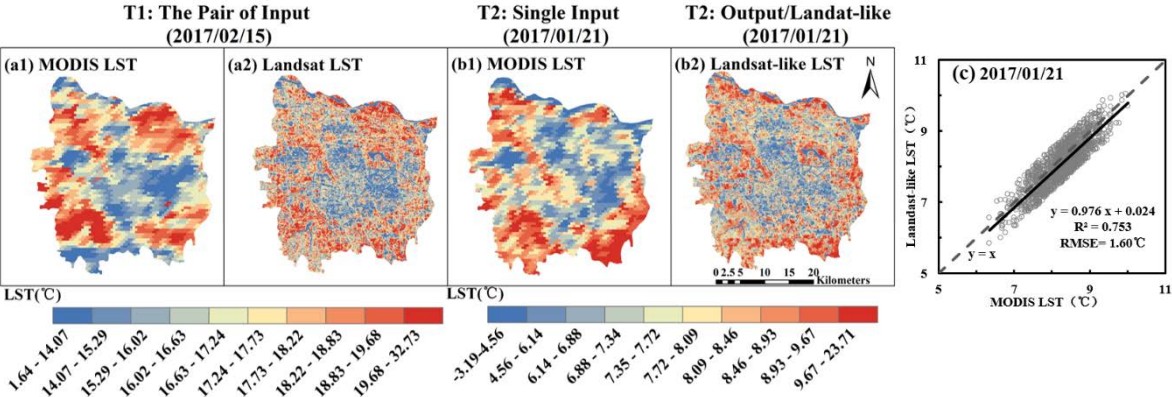

**Figure 4.** Input data for predicting "Landsat-like" LST in 21 January 2017 and accuracy assessment of the output.

As shown in Table 3, for various LULC types, the RMSEs between the Landsat-like LST and the Landsat LST data were also mostly higher than those between the Landsat-like LST and the MODIS LST data except for the RMSE of water on 28 June 2013. The RMSEs in summer were higher than those

in winter. The applicability of the FSDAF method in generating high spatial-temporal-resolution LST is greatly affected by different LULC effects [48,60].

**Table 3.** The root-mean-square error RMSE (°C) of different land use/land cover (LULC) between Landsat/MODIS LST and "Landsat-like" LST.

| LULC Type | 28 June 2013 | | 22 January 2014 | |
| --- | --- | --- | --- | --- |
| | **Landsat LST** | **MODIS LST** | **Landsat LST** | **MODIS LST** |
| ISA | 2.25 | 1.73 | 1.74 | 1.33 |
| Vegetation | 2.70 | 2.42 | 1.69 | 1.21 |
| Bare Soil | 2.22 | 1.69 | 1.70 | 1.27 |
| Water | 2.76 | 2.87 | 1.65 | 1.28 |

ISA: impervious surface area.

### 3.2. LST Variations under Urban Expansion

Urban expansions and land use/land cover (LULC) changes over the period are illustrated in Figures 5 and 6. There were significant seasonal changes and spatial variations in LULC. Urban expansion presented the continual increase of impervious surface area (ISA) in four periods with 389.85, 413.75, 484.56, and 502.69 km$^2$, respectively. The ratio of ISA increased from 37.91% to 40.23%, 47.12%, and 48.88%. The total area of vegetation and bare soil decreased during urban expansion, with an obvious seasonal variation. The overall classification accuracies of the four periods were 89.25%, 88.33%, 91.27% and 94.69%, and the Kappa coefficients were 0.86, 0.88, 0.90, and 0.94 (Table S2).

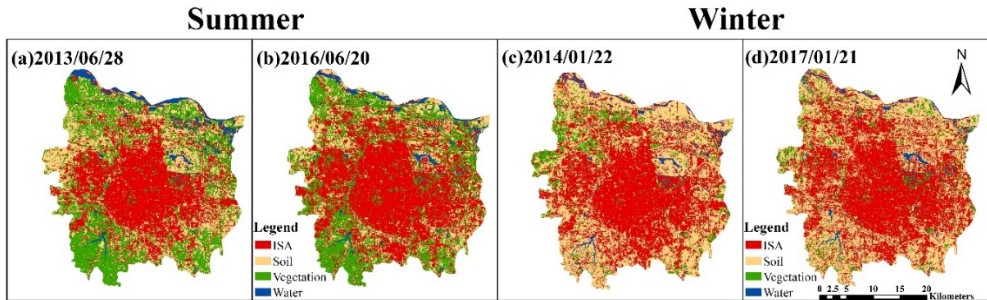

**Figure 5.** Spatial variations of LULC in Zhengzhou on four dates.

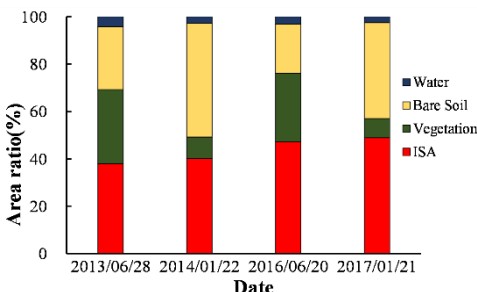

**Figure 6.** Land use/land cover changes in four periods.

In summer, the relatively higher LST (more than 50 °C) regions were mainly concentrated in the center of the study area, where the highest LST on 20 June 2016 was up to 61.89 °C, while the air temperature was 30.4 °C that day. In winter, the LST of the central urban area was significantly lower than that of the surrounding suburbs (Figure 7), and the lowest LST was −3.92 °C on 22 January 2014, while the air temperature was 2.5 °C. It is clear that the LST of most regions in summer of 2016 was higher than that of 2013, while for winter, the differences were not significant. Although the

heterogeneity of the land surface causes the varieties of LST spatially, LSTs are mainly dominated by the distribution of LULC. In summer, the lower LST regions were mainly distributed in the southwest and the north covered by water bodies and grass, while in winter, the higher LST characteristics were more prominent in the surrounding suburbs, while the lower LST zones mostly appeared in the central urban area. Comparing the spatial distribution changes in LST between 2013 and 2016, we found the higher LST regions in summer increased significantly, showing a trend of expansion towards the southwest, while the LST also increased slightly in the west and the southeast of the study area.

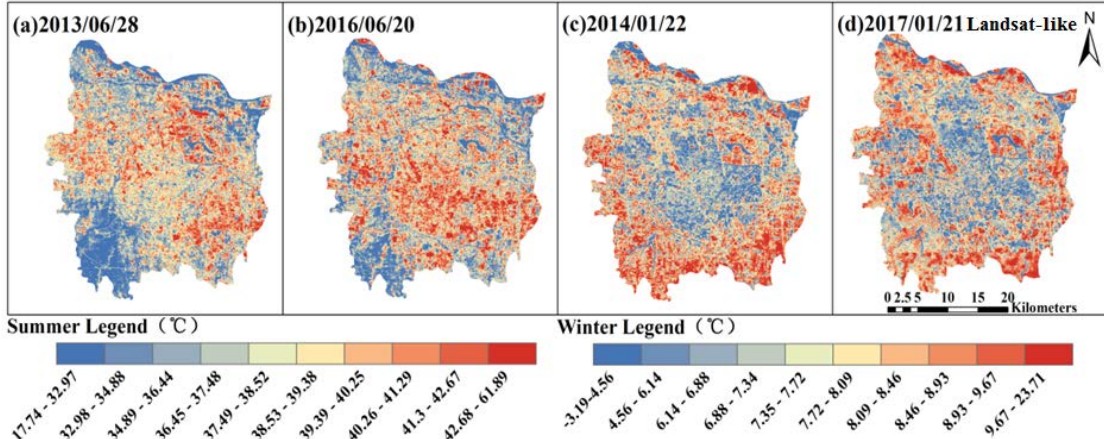

**Figure 7.** LST distribution in four periods.

To reveal the LST spatial variations, the mean value in five different municipal zones of Zhengzhou was analyzed (Figure 8a). From 2013 to 2016, the mean LST in summer changed the most by increasing 2.25 °C in the EQ region compared with changes in other regions (JS −0.05 °C, GC 0.44 °C, ZY 0.68 °C, HJ 0.48 °C), while in winter, the mean LST decreased in all zones. ISA ratio continued to increase over time in each municipal zone and showed the urban expansion mainly depending on the built area increase (Figure 8b). The mean values of NDVI and NDBI also changed (Figure 8c,d). In the EQ region with a big park and green land, the mean NDVI was the highest, while the NDBI was the smallest in summer. The NDBI in winter seasons was larger than that in summer because of the withering vegetation in grassland and few crops covering some farmlands. By comparing the changes in two periods, both the maximum increase of NDBI and the maximum decrease of NDVI in summer were identified in EQ region. The spatial distributions of LST and three indices (including ISA ratio, mean NDVI, mean NDBI) indicate an obvious consistent or opposite trend change for each municipal zone in summer.

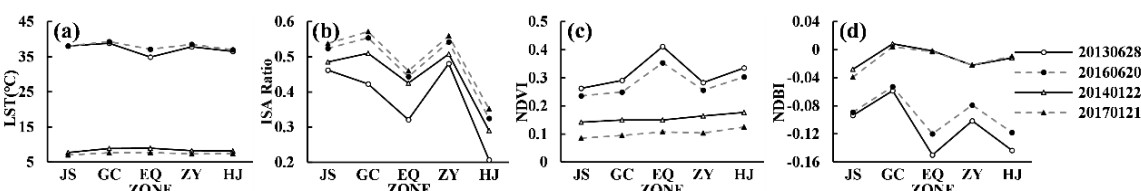

**Figure 8.** Mean value of LST (**a**), ISA ratio (**b**), Normalized Difference Vegetation Index (NDVI) (**c**), and Normalized Difference Built Index (NDBI) (**d**) in different zones.

Figure 9 shows the mean LST in different LULC types by using random points (n = 3000). The LST orders of different LULC in four periods were listed as follows: bare soil > ISA > vegetation > water (28 June 2013), ISA > bare soil > vegetation > water (20 June 2016), and bare soil > vegetation > ISA > water (28 June 2013 and 21 January 2017). In summer, ISA (including industrial lands, residential areas, and commercial regions) and bare soil had relatively higher LST than vegetation and water because of the lower specific heat capacities.

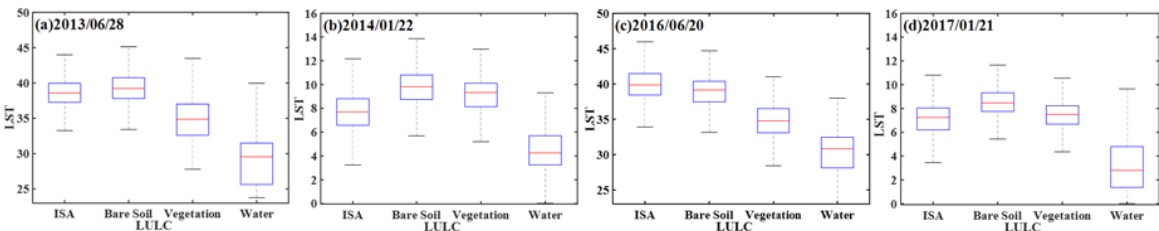

**Figure 9.** Box graphs of the relationship between LULC and LST (unit: °C).

Urban expansion includes not only the total area increase but also internal urban land increase. Figure 10 shows spatially the ISA changes and the corresponding LST differences between two summers of 2013 and 2016. The ISA increase regions existed in various municipal zones, especially in the southwest of JS, the northwest of GC, the northeast of EQ, and the southeast of ZY (Figure 10a). The LST increase regions mostly occurred in the area where the LST decreased (Figure 10b).

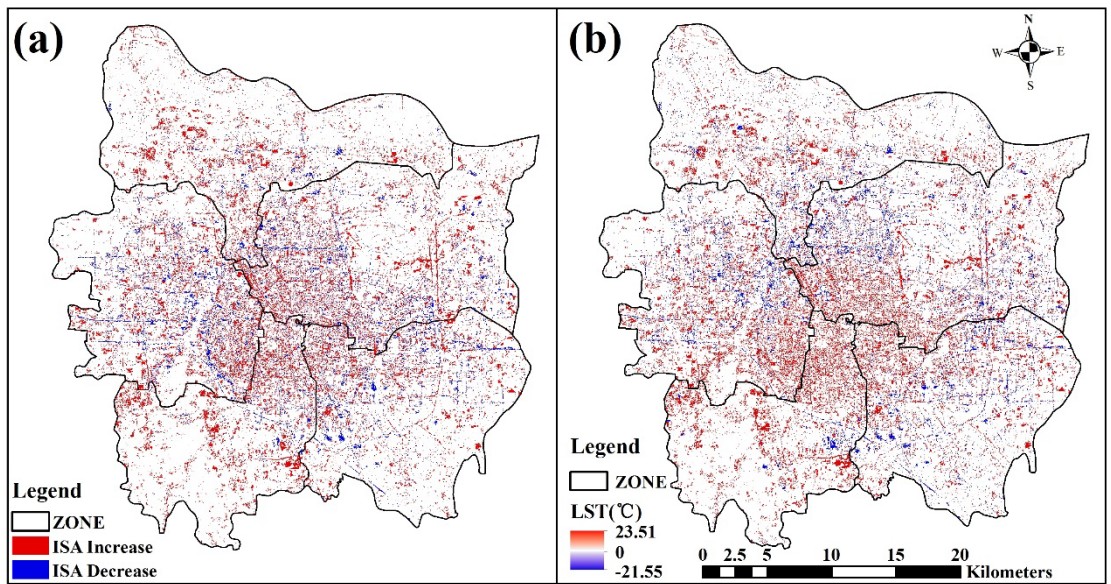

**Figure 10.** ISA changes and corresponding LST difference between 28 June 2013 and 20 June 2016. (**a**) The ISA regional changes generated by using the NDBI threshold method; and (**b**) the spatial distribution of LST difference.

As shown in Figure 11a, the mean LST differences of two periods were calculated for parts where ISA changed or not in each zone. For ISA increase type, LST increased with the range 1.9 to 3.5 °C, showing a "warming effect" of the building land. LST decreased with the range −3.8 to −0.5 °C where ISA decreased. The mean LST difference presented a small increase and ranged from −0.15 °C to 1.98 °C where ISA had no change. By comparing two term changes in EQ zone, the largest LST increased 3.5 °C for the parts where ISA increased. LST increased where other types of LULC were converted to ISA; on the contrary, LST decreased where ISA regions were converted to non-ISA regions. As shown in Figure 11b,c, including ISA increased region and ISA decreased region, a strong nonlinear quadratic polynomial correlation between ISA ratio change and LST change existed ($R^2$s were 0.698 and 0.652, respectively, $p < 0.001$). As for the ISA increased areas, LST showed 0.27 °C growth as ISA ratio increased by 0.1. However, for ISA decreased regions, LST showed two different trends. LST increased by 0.25 °C as ISA ratio increased by 0.1 in the range from −1 to −0.3, while LST decreased by 0.22 °C as ISA ratio increased by 0.1 in the range from −0.3 to 0.

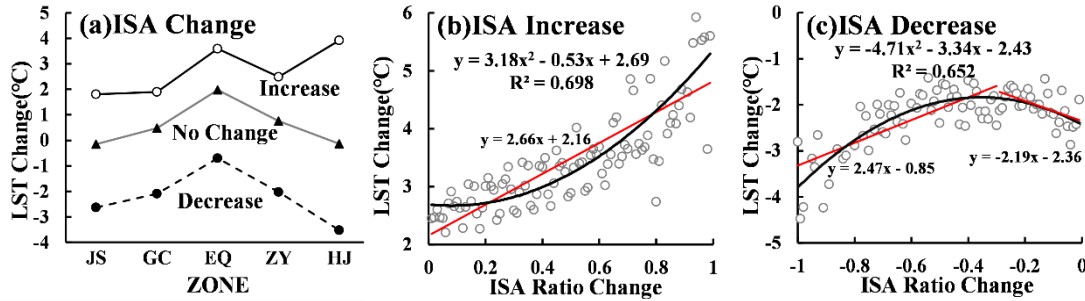

**Figure 11.** The impacts of ISA changes on LST changes.

### 3.3. Driving Factors on LST

NDVI and NDBI are two important factors reflecting the LULC change. Figure 12 shows correlations of NDVI-LST and NDBI-LST to reveal the driving forces (mainly focusing on vegetation and ISA) on LST. LST had a negative linear relationship with NDVI ($R^2 = 0.425$ and $R^2 = 0.549$, $p < 0.001$), whereas there was a less obvious linear relationship in winter. NDBI had a positive relationship with LST that was relatively stronger in summer ($R^2 = 0.601$ and $0.609$) and weaker in winter ($R^2 = 0.308$ and $0.214$). The NDBI had a closer relationship with LST than NDVI, further illustrating that the building land increase had a more significant impact on LST than vegetation coverage during urban expansion.

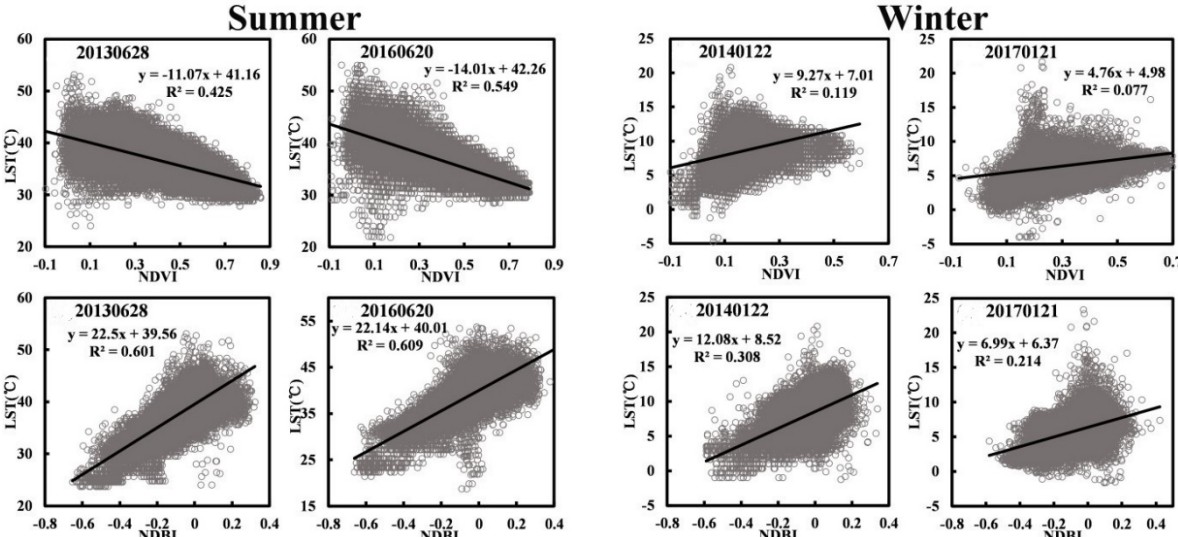

**Figure 12.** The correlation of LST with NDVI and NDBI in summer and winter.

### 3.4. Integrated Monthly LST Dynamics Based on Landsat-like Data

For generating monthly "Landsat-like" LST using MODIS 11A2 products and Landsat LST, four 8-day "Landsat-like" LST data in each month were generated to integrate the monthly LST with maximum value composite method. For example, the first two predicated LST (Figure 13a$_1$,a$_2$) and last two "Landsat-like" LST (Figure 13a$_3$,a$_4$) of June were generated by using the Land satellite images of 12 June 2013 and 28 June 2013 individually, while predicated LST (Figure 13b$_1$–b$_4$) were all generated by using the Landsat LST image on 13 December 2013.

As shown in Figure 14, with the available Landsat images of each month (Table 4), the integrated monthly "Landsat-like" LST of 2013 were generated. The continuous high resolution monthly LST data can provide sufficient information to illustrate the serial LST dynamics in the regional thermal environment over the long-term and with high precision. The monthly LST values of different percentiles in 2013 were calculated, and the distribution trend is shown as Figure 15. From January to December,

distinct seasonal variations showed the highest LST (36.45 °C) in summer and the lowest LST (9.29 °C) in winter.

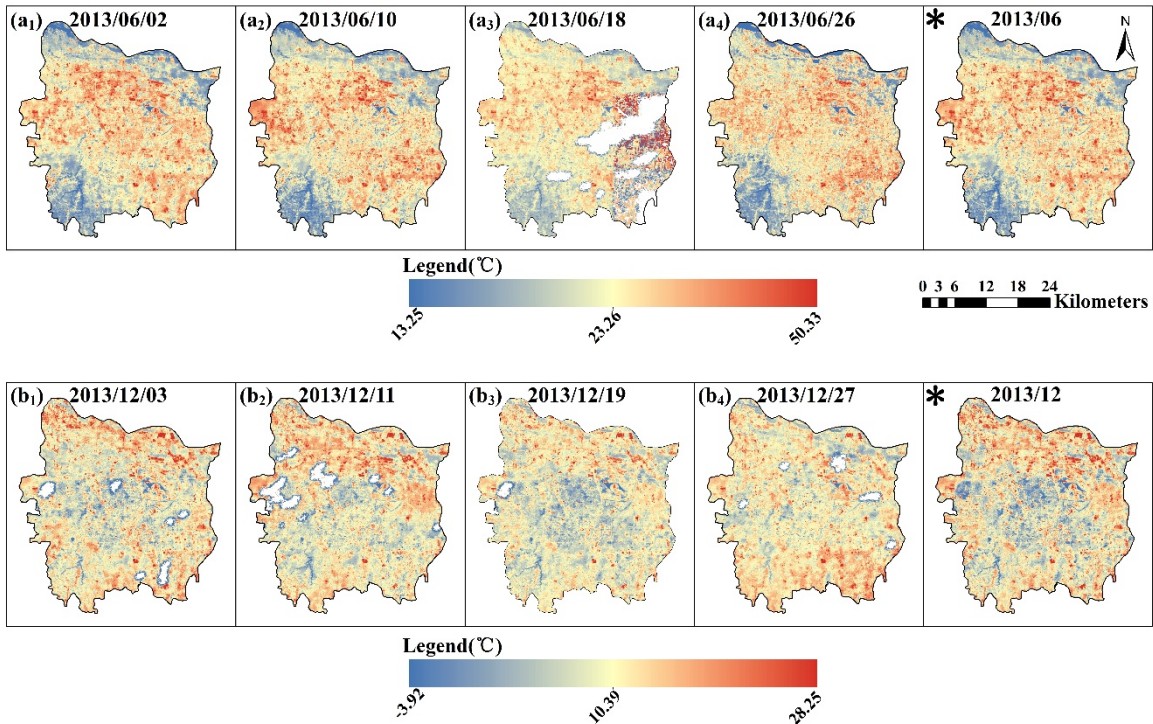

**Figure 13.** The monthly "Landsat-like" LST of June and December in 2013. Two rightmost (*) represent the integrated monthly "Landsat-like" LST. (a_1)–(a_4) and (b_1)–(b_4) mean 8-day "Landsat-like" LST.

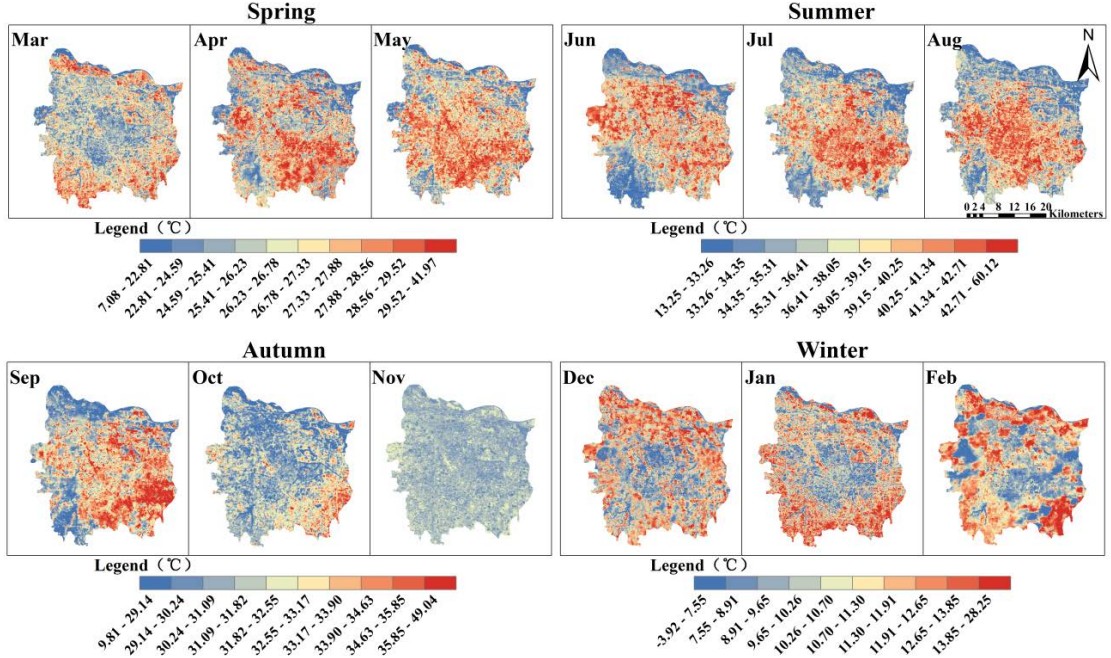

**Figure 14.** The distribution of monthly "Landsat-like" LSTs in 2013.

**Table 4.** The number of available Landsat images in each month of 2013.

| The Number of Available Landsat Images | Month |
|---|---|
| 0 | April, July, and September of 2013, February of 2014 |
| 1 | November and December of 2013, January of 2014 |
| 2 | March, May, June, August, and October of 2013 |

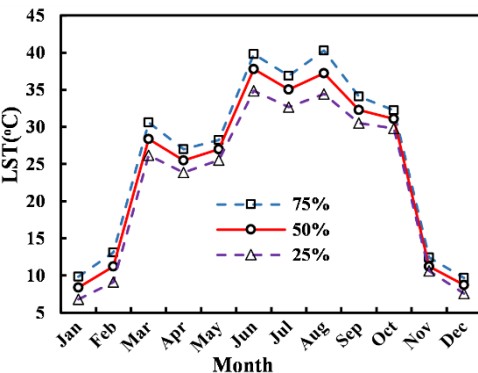

**Figure 15.** The monthly "Landsat-like" LSTs with 75%, 50%, and 25% percentiles.

The relationship of NDVI-LST and NDBI-LST is illustrated in Figure S1. The correlation coefficient $R^2$s of NDVI-LST and NDBI-LST in summer were above 0.4 (June, July, and August of 2013), while $R^2$ was rather weak in winter (Figure 16). From the monthly series of correlation analysis, the positive correlation of NDBI-LST was more stable and stronger than the negative relationship of NDVI-LST. At the same time, the surface parameter NDVI had the largest driving force on LST. The seasonal difference in the correlation of NDVI-LST was significant, especially among summer and the other seasons, while the correlation of NDBI-LST only presented the lowest in winter and remained stably strong during the year of 2013.

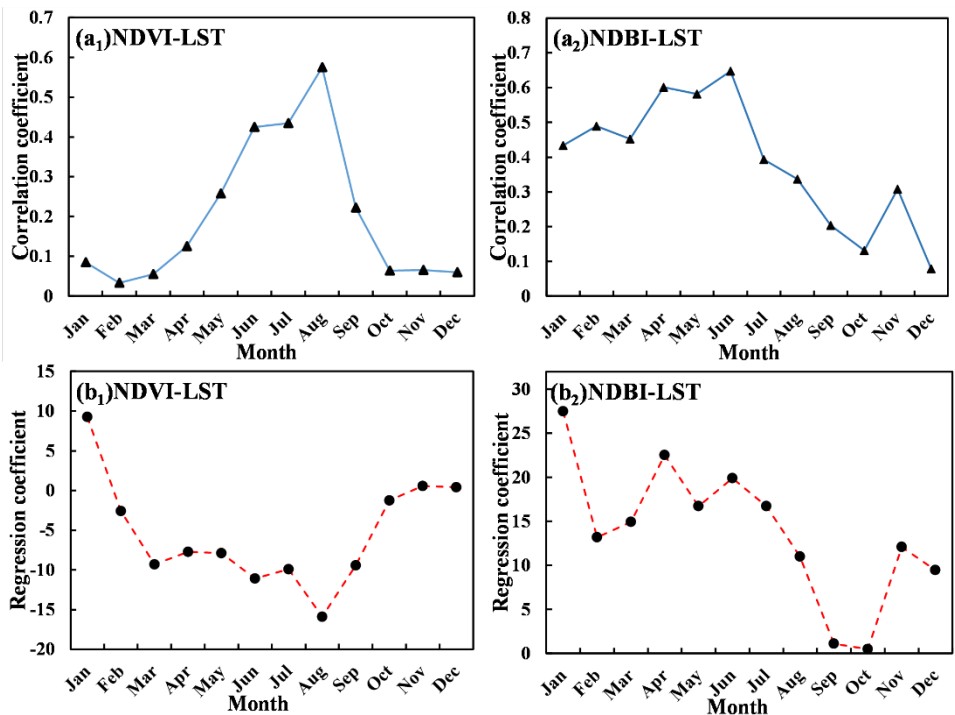

**Figure 16.** Relationships between Landsat NDVI, NDBI, and LST from monthly series of 2013.

As shown in Figure 17, air temperature exhibited a similar monthly trend to LST ($R^2$ = 0.878). In August, the highest monthly air temperature was 30 °C, while the highest monthly LST was up to an average 37.44 °C for the whole city. However, the highest pixel LST was more than 50 °C, which was almost 13 °C higher than the air temperature in August. A moderate negative correlation ($R^2$ = 0.696) was detected between particulate matter PM2.5 and LST. Generally, PM2.5 can reduce the amount of surface solar irradiances reaching the ground surface and cause a cooling effect on the surface. In contrast, there were no strong results showing effects of wind speed and rain on LST variations.

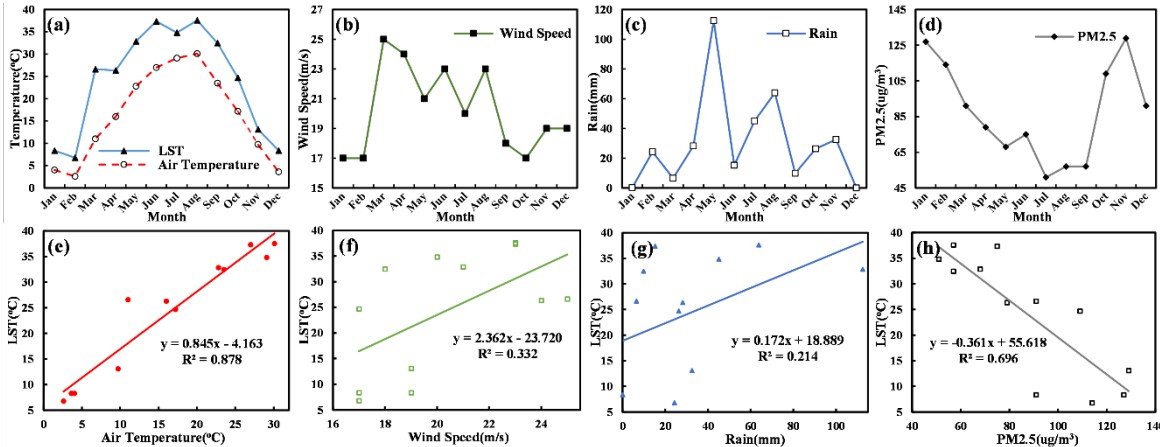

**Figure 17.** (**a**)–(**d**) represent monthly series trend of LST and air temperature, wind speed, rain, and PM2.5; (**e**)–(**h**) represent correlation of air temperature-LST, wind speed-LST, rain-LST, and PM2.5-LST, respectively.

## 4. Discussion

Spatiotemporal fusion models can make up the limitations of remotely sensed data missing or cloud contamination in the related thermal environment researches, and they provide a new method for previous researchers who had to use single daily LST data representing continuous data, because of the satellites' long return time cycle [50,61]. Especially, the flexibility and the performance of the FSDAF method has higher efficiency and lower input requirement. Compared with the results of previous studies [48], the RMSE between predicted LST and the observed LST in this study could be kept within 2.50 °C, showing a high precision level. However, the result of Landsat-like LST is dependent on the date of input Landsat image acquisition and is especially influenced by using the replacement data because of data missing or cloud contamination. Figure 18 shows the closer the input Landsat image acquisition date to the predicted date is, the higher the accuracy of the result is. By selecting Landsat LST of 28 June 2013 as the reference, "Landsat-like" LST predicted 4 June 2013, which was the closest to the output predicted date showing the smallest differences from actual LST [Figure 18b]. Comparatively, LST differences greater or less than 3 °C were more obvious if using Landsat-like LST generated from 11 May 2013 [Figure 18a] or 31 August 2013 [Figure 18c]. Also, the selection date of the paired input data also affects the predicted Landsat LST. The input coarse MODIS LSTs of various dates have differences in the satellite viewing angles and sun geometry, among others, which also cause some deviations between actual LST and predicted LST. As to the serial LST dynamics analysis, 8-day Landsat-like LSTs were predicated and then were combined to generate the integrated monthly Landsat-like LST by using the maximum value composite method. However, due to the cloud coverage influence, uncertainties and new methods should be solved while using the FSDAF method for large scale LST analysis [62,63].

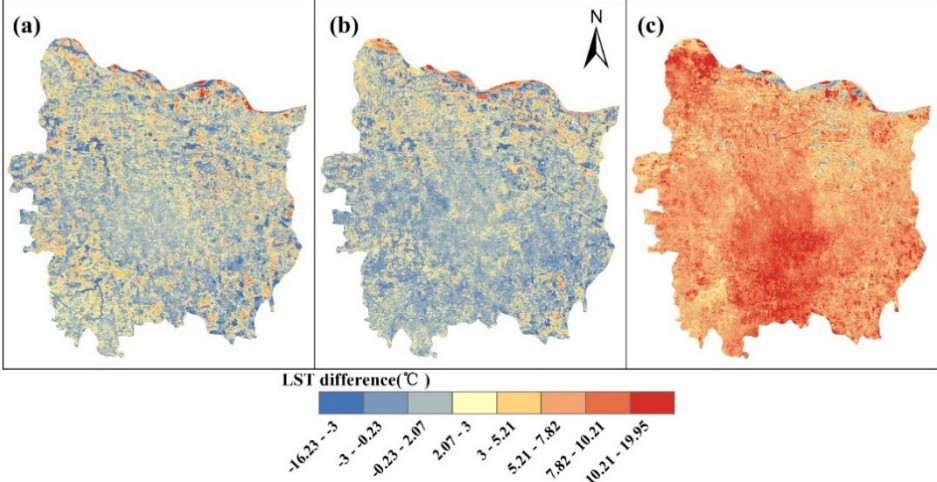

**Figure 18.** Difference between Landsat LST (28 June 2013) and three predicted "Landsat-like" LST generated by using one single input MODIS LST (28 June 2013) and three different pairs of inputs (MODIS LST and Landsat LST) on different dates: 11 May 2013 (**a**), 4 June 2013 (**b**), and 31 August 2013 (**c**).

The urban area of Zhengzhou city expanded by 28.94% from 2013 to 2016. This is similar to previous studies that revealed a continuous and rapid expansion in the process of Zhengzhou urbanization for many years [64,65]. The urban expansion affected LST with great variations. In summer, the center of the urban area had a relatively higher LST than the surrounding suburbs, while it presented a completely opposite LST distribution in winter. This phenomenon is widespread in most major cities in north and central-north China. In Zhou's research, spatial distributions of surface Urban Heat Island Intensity (SUHII) of Zhengzhou ranged from 2 °C to 3 °C in summer and from −0.5 °C to 0 °C in winter [32]. Yao et al. also proposed that the SUHII of Zhengzhou ranged from 2 °C to 4 °C in summer and from −2 °C to 0 °C in winter [40]. However, more attention should be paid to explain uncertainties about the urban cooling effect, especially by combining the results of the numerical simulation and the remote sensing retrieval method. As for the driving forces on LST, similarly to other research [66], in summer months, NDVI-LST showed a negative correlation, while NDBI had a quite strongly positive correlation with LST. However, due to the seasonal fluctuations of vegetation, the relationship of NDVI-LST changed dramatically and showed weaker correlations in winter, as reported in other research [67,68]. Different LULC types had different specific heat capacities, and vegetation was transferred to built-up land, which increased LST [69]. This is consistent with the results of previous studies [70], indicating that ISA changes have a large impact on the spatial distribution of LST.

In addition, except for ISA, NDVI, and NDBI, a range of climatic factors including air temperature, wind speed, and rain can also affect the LST variations [23]. Research has disclosed climatic multiple effects on LST and pointed out the potential strategies for heat stress alleviation and urban planning [71,72]. For example, the strong correlation between air temperature and LST could be used to perform estimation of spatially distributed near surface air temperature with constructing models [73,74]. However, according to Figure 17a, we can see the biggest difference between LST and air temperature was 15 °C in March. The reason for this may have been the fast vegetation growth inducing the LST increase. In the whole year, LST fluctuated in some months while there was only one peak in the air temperature curve. That shows the complex thermal variations mechanics of land surface in respect to radiance and emission. Although we pointed out the negative impacts of PM2.5 on LST, there are still great uncertainties. For example, Cao et al. pointed out urban haze/aerosol pollution was also a great contributor that intensifies the high surface temperature agglomeration effect on the center of the urban area at night [75]. The wind effects may differ according to different location and topography. The change of the urban boundary layer wind field reduces the LST of the urban center, especially in coastal cities, and good ventilation can better improve the effects of the urban thermal environment [76]. With long-term and multi-site data, it could

be disclosed that there is a negative correlation presented between LST and precipitation [73]. In addition to these significant factors that have a great impact on the LST changes, there are still some possible contributors, e.g., incoming surface radiation, duration and intensity of sunlight, elevation, etc. [77]. LST variations are affected by the contribution from multi factors, and comprehensive analysis is helpful for fully understanding the relevant driving forces and driving modes [78].

## 5. Conclusions

In this study, to overcome the shortage of coarser spatial resolution of MODIS data and lower temporal resolution of Landsat data, "Landsat-like" datasets were generated using the FSDAF method, including LST, NDVI, and NDBI. Under the urban expansion, LST variations and related driving factors were analyzed.

The results show that high spatiotemporal-resolution "Landsat-like" LST has a high accuracy level, and the overall trend is consistent with Landsat LST and MODIS LST for the same date. From July 2013 to July 2016, the urban area expanded from 389.85 km$^2$ to 484.56 km$^2$. LST increases ranging from 1.80 °C to 3.92 °C were detected in areas where the impervious surface area increased, while LST decreases ranging from −3.52 °C to −0.70 °C were detected in areas where ISA decreased. The relationship between NDVI and LST showed significant seasonal differences; in summer, it revealed a negative linear relationship ($R^2 = 0.425$ and $R^2 = 0.549$, $p < 0.001$). NDBI presented a strongly positive correlation with LST in summer with $R^2$ more than 0.6. Monthly Landsat-like LSTs were generated from four predicated 8-day "Landsat-like" LST in each month to indicate comprehensive dynamics. Also, monthly Landsat-like NDVI and NDBI, air temperature, PM2.5, and other factors were analyzed to disclose the driving forces. In future research, temporal and spatial fusion efficiency and precision can be improved by using multi-source data or deep-learning algorithms. Also, urbanization effects on thermal environment should be analyzed by coupling different dimensional data including land surface factors, energy consumption, and urban morphology. This study can provide practical information and guidance to assist local thermal environmental management and to advise decision-makers about good land use practices to be applied during urban expansion.

**Supplementary Materials:** The following are available online at http://www.mdpi.com/2072-4292/12/5/801/s1: Remote sensing datasets, **Table S1:** The monthly series about the correlations of LST with NDVI and NDBI in 2013, **Table S2:** Accuracy assessment of Land Use/Land Cover (LULC) classification.

**Author Contributions:** Conceptualization, H.Y. and Z.W.; Investigation, C.X.; Methodology, P.M. and X.Z.; Validation, Y.S. and T.H.; Writing – original draft, H.Y. and P.M.; Writing – review & editing, X.Z. and Z.L. All authors have read and agreed to the published version of the manuscript.

**Funding:** This research was supported by Henan Province Scientific and Technological Project (Grant Nos. 162102410066 & 172102410075). Key scientific research projects of Henan colleges and universities (Grant Nos. 19A170014 &18A170014). The work was also supported by the UK Science and Technology Facilities Council (STFC) through the PAFiC project (ref.: ST/N006801/1).

**Acknowledgments:** This authors would like to thank National Aeronautics and Space Administration for providing us with the MODIS data (ftp://ladsweb.nascom.nasa.gov).

**Conflicts of Interest:** The authors declare no conflict of interest.

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
