# Peer review of "Measuring the Urban Land Surface Temperature Variations Under Zhengzhou City Expansion Using Landsat-Like Data"

_remotesensing, doi:10.3390/rs12050801_

Round 1

Reviewer 1 Report

Dear authors,

The topic of your manuscript is quite interesting since it focuses on the LST of urban expansion areas in Zhengzhou City of China, combining MODIS and Landsat data. Please consider the following comments in order to improve the quality of the manuscript.

General Comment: Although the aim of this study is interesting you did not manage to present many meaningful results. Although the main research question is clear (What is the effect on LST in the urban expansion areas) the final results can barely provide an advance in the current scientific knowledge. You should further explore the findings of your analysis in all the ways you have already mentioned in your manuscript. Despite the fact that you present many results an overall outcome is not clear. The methodology used is already implemented by others; the advantage of implementing this methodology in the city of Zhengzhou is not supported by the results.

Specific comment 1: Please enhance your results throughout manuscript with physical meaning and explanation, so as to improve your scientific analysis and conclusions. For example lines 251-269 a result of all this analysis is missing. Does ISA increase leads to LST increase? What about an increase in NDVI?

Specific comment 2: Fusion analysis and methodology is not utilized; the analysis is presented in a coarser scale, in municipal zones. Why employing fusion techniques and then analyzing data in coarser spatial resolution? (See also next comment)

Specific comment 3: In some parts of the manuscript you provide only qualitative analysis. For example lines 238-248 quantitative results are missing; you present just interpretation of the figures without quantitative results. Since you have the LULC classes, why don’t you present LST per class? In this way you can utilize the fusion methodology.

Specific comment 4: The expansion percentage of the Zhengzhou city is presented only in the discussion section (line 371). Why don’t you mention it also in the corresponding section 3.2. Furthermore, Figure 4 can be replaced with a Figure showing only the differences (changes) in LULC between the years, in that way city expansion will be clearly presented.  

Specific comment 5: Why to generate monthly LST? What is the need and how it is connected with the main aim of your study? LULC changes do not occur in monthly base. Can driving forces of the LST be determined from the daily data?  

Specific comment 6: Is one meteorological stations representative for the entire area?

Specific comment 6: Lines 383 -397 discuss results only from other studies, which are the corresponding results of your analysis?

Reviewer 2 Report

This paper is to propose a methodology to measure urban land surface temperature variations under the four periods of urban expansion by using Landsat-like LST and also analyzing LST driving factors to identify LST dynamics. This topic is relevant to the journal. However, various points need revisions taking into account the following comments:

In the introduction, you need to connect the state of the art to your paper goals. Please follow the literature review by a clear and concise state of the art analysis. This should clearly show the knowledge gaps identified and link them to your paper goals. Please reason both the novelty and the relevance of your paper goals. The methodology is clear and very well presented but it is necessary to incorporate some minor revisions. This section should be completed with some relevant information and a more complex discussion would be needed.  Recommendations for future studies are not significant as per the findings of this study, so rewrite the future studies Conclusions should be improved by including more information (data) on the analyzed results. Keywords should not be the repetitions of the title words, please find such words which are not in the title, this way search engines of the web will find your manuscript with higher probability.

Reviewer 3 Report

Overall impression:

In this paper, the authors introduce a flexible spatiotemporal data fusion method for upscaling thermal satellite images and bring the example of usefulness on investigation Zhengzhou City evolution of thermal conditions. The overall impression of the text is good. Paper has a good structure and, it is well written and easy to read. The paper has two parts. The first methodologic part about FSDAF is still novel and in my opinion interesting to the world audience. The second part about the thermal conditions of city Zhengzhou is more case-specific and it is difficult to generalize or apply the results to other cities in different parts of the world and different climate conditions.

Specific Comments:

Lines 52/53: impervious surface cover (ISC) is used but in the rest of the paper you are using ISA. If they are the same please choose just one therm.

Line 80: Typo M7oreover.

Line 90: Expanding of the area needs a reference.

Line 91: Typo (Table S1).

Lines 106 – 120: This part is quite confusing because of the mixed pre-processing of thermal and optical data. I.E. Line 111: FLAASH can’t be applied on thermal data, Line 110: Radiance units are useful for thermal data but for optical data reflectance values should be used, etc…

Line 106, 109: Did you manually orthorectified MODIS and Landsat data? The satellite images are already orthorectified by a provider.

Line 108: Which resampling method was used?

Lines 153 – 164: Processing graph could help to understand the workflow.

Line 203: The more appropriate method would be to test minimum distance for avoiding autocorrelation between points and then use random points with minimum distance span. It is just my opinion, you can keep it as it is.

Line 208: Can you compare this RMSE with temporal oscillation? How big is RMSE between images from two close dates of Landsat LST? Is this RMES smaller? My point is: In case I need a thermal image with a fine resolution of the date I don’t have, I have two possibilities. A) Create one with FSDAF from coarse resolution B) Use some image that is close to the desired date. Which possibility will be more correct?

Line 339: PM2.5 should be explained here, but it is explained on line 384.

Round 2

Reviewer 1 Report

Author's reply along with the amendments made on the manuscript based on all reviewers comments improved the quality of the manuscript.

Thank you for your effort and for considering my comments.

Reviewer 2 Report

The authors performed extensive paper revision and enhancement work, addressing most of the suggestions made in the previous review. The paper is considered to have attained a sufficient quality to be considered for publication.

Reviewer 3 Report

Thank you for your reply. The authors fulfilled all my comments. In my opinion, the paper can be published in the present form.